# Structure and Vibrational Spectroscopy of C_82_ Fullerenol Valent Isomers: An Experimental and Theoretical Joint Study

**DOI:** 10.3390/molecules28041569

**Published:** 2023-02-06

**Authors:** Felix N. Tomilin, Polina V. Artyushenko, Irina A. Shchugoreva, Anastasia V. Rogova, Natalia G. Vnukova, Grigory N. Churilov, Nikolay P. Shestakov, Olga N. Tchaikovskaya, Sergei G. Ovchinnikov, Pavel V. Avramov

**Affiliations:** 1Kirensky Institute of Physics, Federal Research Center KSC SB RAS, Krasnoyarsk 660036, Russia; 2School of Non-Ferrous Metals and Materials Science, Siberian Federal University, Krasnoyarsk 660041, Russia; 3Laboratory for Digital Controlled Drugs and Theranostics, Federal Research Center Krasnoyarsk Scientific Center of the Siberian Branch of the RAS, Krasnoyarsk 660036, Russia; 4Laboratory for Biomolecular and Medical Technologies, Prof. V.F. Voino-Yasenetsky Krasnoyarsk State Medical University, Krasnoyarsk 660022, Russia; 5Department of Physics, Tomsk State University, Tomsk 634050, Russia; 6Department of Chemistry, Kyungpook National University, Daegu 41566, Republic of Korea

**Keywords:** C_82_, Gd endohedral complexes, biomedical applications, fullerenols, DFTB3 electronic structure calculations, IR spectra

## Abstract

Gd@C_82_O_x_H_y_ endohedral complexes for advanced biomedical applications (computer tomography, cancer treatment, etc.) were synthesized using high-frequency arc plasma discharge through a mixture of graphite and Gd_2_O_3_ oxide. The Gd@C_82_ endohedral complex was isolated by high-efficiency liquid chromatography and consequently oxidized with the formation of a family of Gd endohedral fullerenols with gross formula Gd@C_82_O_8_(OH)_20_. Fourier-transformed infrared (FTIR) spectroscopy was used to study the structure and spectroscopic properties of the complexes in combination with the DFTB3 electronic structure calculations and infrared spectra simulations. It was shown that the main IR spectral features are formed by a fullerenole C_82_ cage that allows one to consider the force constants at the DFTB3 level of theory without consideration of gadolinium endohedral ions inside the carbon cage. Based on the comparison of experimental FTIR and theoretical DFTB3 IR spectra, it was found that oxidation of the C_82_ cage causes the formation of Gd@C_82_O_28_H_20_, with a breakdown of the integrity of the parent C_82_ cage with the formation of pores between neighboring carbonyl and carboxyl groups. The Gd@C_82_O_6_(OOH)_2_(OH)_18_ endohedral complex with epoxy, carbonyl and carboxyl groups was considered the most reliable fullerenole structural model.

## 1. Introduction

Water-soluble fullerene derivatives attract considerable attention because of their potential for a variety of advanced biomedical applications. In particular, fullerenes reveal antiviral activity [1,2,3]. Shoji et. al. [2] found that C_60_ fullerene derivatives inhibit influenza A viral infection and can be considered as prospective candidates for the development of anti-influenza drugs. Water-soluble derivatives of fullerene C_60_ show a potential as biological antioxidants due to their chemically functionalized external surfaces by hydrophobic groups such as −OH and −COOH [4,5,6]. Fullerene derivatives can serve as drug delivery scaffolds with covalent or noncovalent linkages between the carbon cage and a bioactive moiety [7,8,9]. Highly iodinated fullerene derivatives [10] are of interest as potential imaging agents due to the high radioactive contrast of iodine ions. Gadolinium endohedral complexes of fullerenols are considered as prospective magnetic resonance imaging contrast agents [11,12,13,14,15]. Fullerene derivatives conjugated with single-stranded oligonucleotides called aptamers are considered a promising drug for cancer treatment [16] and several other diseases. The almost complete insolubility of pure fullerenes can be overcome by the attachment of various polar functional groups to the fullerene cages [17,18,19]. The presence and mutual arrangement of different functional groups affect the electronic structure, chemical activity and optical properties of fullerene derivatives [20,21,22].

Despite the desperate demand caused by a wide range of possible applications of such types of compounds in medicine, determination of the exact atomic structure of fullerene derivatives is still a challenging job. Density functional theory (DFT) calculations are a powerful tool to describe atomic and electronic structures and the stability of different derivatives of fullerenes [23,24,25,26], including endohedral metallofullerenes [27,28,29,30,31,32]. A joint experimental and theoretical approach proved to be a prospective method for the determination of substitution functional groups and their positions on fullerene cages, based on the comparison of experimental IR spectra of fullerene derivatives with theoretical ones. The dependence of the IR spectra upon the distribution of functional groups was successfully studied using DFT electronic structure calculations of hydroxylated [26,33] and oxidized [34] C_60_ derivatives. Since the vibration frequencies of the carbon cage and oxygen–containing groups lie in the range of 1000–4000 cm^−1^, this spectral energy region of fullerene derivatives is of special interest when determining the atomic structure of fullerenols by comparing the experimental and theoretical IR spectra [14,23]. For endohedral fullerenols, the guest metallic ions absorption bands do not fall into this region and do not need to be considered [14,35].

Among all types of fullerenes and their derivatives, with the number of carbon atoms in the parent carbon shells ranging from 16 to the hundreds [36], the C_60_ derivatives and complexes are the most studied [1,2,4,5,10,11,12,18] to date. However, of all endohedral fullerenes known so far, the C_82_ fullerene has become one of the most intensively studied cages [37,38,39]. Hydroxylation of a carbon cage is one of the common strategies used to solubilize the endohedral Gd@C_82_ complex [40]. Although considerable research has been devoted to the medical application of Gd@C_82_ and its derivatives [41,42,43], rather less attention has been paid to the determination of the atomic structure of functionalized Gd@C_82_ fullerene [22,44,45]. At the moment, the description of the atomic structure of fullerene is based on the prediction of the stability of various models by the method of density functional theory without comparisons with available experimental structural and spectroscopic data [22,44,45].

The Gd@C_82_O_x_H_y_ complexes were successfully synthesized earlier [46], but their atomic structure has not been properly characterized yet. In a previous work [22], a number of atomistic models of endohedral fullerenols Gd@C_82_O_x_H_y_ (x = 0 and 3; y = 8, 16, 24, 36 and 44) were proposed and studied using a combination of experimental mass spectroscopy techniques and density functional theory calculations. As a result of the combined experimental and theoretical investigation, the endohedral Gd@C_82_O_x_H_y_ complex with 24 hydroxyl groups was considered the most prospective candidate for biomedical applications [22].

This study is devoted to the harsh–condition synthesis and joint experimental Fourier transform infrared (FTIR) absorption spectroscopy and the theoretical Density Functional Tight Binding (DFTB3) investigation of various Gd@C_82_O_x_H_y_ complexes. The FTIR spectroscopy revealed an unexpected absorption band in the energy range from 1600 to 1800 cm^−1^ that could be assigned to double carbon–oxygen chemical bonds. For the sake of the proper interpretation of experimental FTIR data, a DFTB3 theoretical study of atomic and electronic structures and optical spectra of Gd@C_82_O_x_H_y_ endohedral complexes was performed using the Density Functional Tight Binding approach. The structure of the proposed theoretical model of Gd@C_82_O_x_H_y_ was found to be in good agreement with the experimental spectroscopic data. As a result, the influence of the key geometric parameters, the types of substitutional functional groups and their number in the fullerene cage were revealed by the DFTB3 method.

## 2. Results and Discussion

The harsh–condition synthesis of water–soluble fullerenols [46] is carried out under rather stringent conditions. This leads to a wide range of fullerenol molecules with different numbers and combinations of oxygen–containing functional groups. This, in turn, leads to difficulties in the determination of the atomic structure by means of X–ray electron diffraction. Therefore, various indirect methods should be and are used to verify the structures, which cannot give a direct and accurate answer about the atomic structure of the fullerenols.

It is reasonable to use infrared spectroscopy and X–ray photoelectron spectroscopy (XPS) to certify the water–soluble fullerenes. The FTIR spectroscopy method allows one to determine the functional groups, while XPS provides the opportunity to calculate the number of bond species of carbon and oxygen atoms. In the XPS fullerenol spectrum, the C_1s_ bond energies extend to 5.3 eV and clearly deviate from the C_1s_ peak of fullerene [47]. The large shift indicates a relatively high percentage of carbon with a higher degree of oxidation in fullerenol compared to fullerene. The wide range and spectral width also directly indicate that a significant number of isomers may be presented in the experimental mixture of the products. An analysis of the data provides the unique possibility to calculate the approximate atomic concentrations of different carbon and oxygen species with different bond types and oxidative states.

In the study, FTIR spectroscopy was used to determine the nature and coordination of the functional groups, since XPS does not provide detailed information about the diversity of functional groups. Certain bands in the experimental FTIR spectrum suggest that the C=O carbonyl, epoxy, −OH hydroxyl and −COOH carboxyl groups can be formed and coordinated to the surface of the fullerenol C_82_ carbon cage [47,48,49]. The atomistic theoretical models of possible C_82_ fullerenol isomers should take into account that the integrity of the parent fullerene cage could be broken by the appearance of C=O carbonyl and -OOH carboxyl groups coordinated to the surface of the molecule. Therefore, it is important to control that the fullerenol molecule does not break down, i.e., does not disintegrate into different species and fragments during geometry optimization in electronic structure calculations.

The approximate number of the functional groups was determined using XPS based on the proportion of carbon atoms chemically bonded to oxygen. Since the number of -OH hydroxyl groups attached to the fullerene must be even, the composition of a given fullerenol can be expressed as Gd@C_82_O_x_H_y_, where x:y ~ 1.5, so, in this work, the molecules C_82_O_6_(OOH)_2_(OH)_18_ and C_82_O_8_(OH)_20_ were constructed with an O:H ratio = 28:20. Since the main comparison of the theoretical models and experimental structures was done using infrared spectroscopy, the main goal in this work was to achieve the correct ratio of functional groups as it appears in the intensity of the corresponding spectral FTIR peaks. At the same time, gadolinium can be neglected in the calculation of IR spectra, since the concentration of one atom is very small compared to the rest, and, which could be even much more important, due to the huge Gadolinium atomic mass; the vibrations of gadolinium ion are in the long wavelength region of the spectrum, and its characteristic lines are poorly or not visible at all on the experimental spectrum. Therefore, the use of simpler methods such as the effective DFTB3 is more reasonable and faster.

### 2.1. FTIR Spectrum of Hydroxylated Gd Endohedral Complexes

The hydroxylated fullerene Gd endohedral complex Gd@C_82_ was synthesized and studied using FTIR spectroscopy. According to Reference [49], absorption bands from C–C (1554 cm^−1^), C=C (1621 cm^–1^), C=O (1703 cm^−1^) and C–O (1050–1150 cm^−1^) bonds were identified. The obtained experimental IR spectrum is presented in Figure 1 (black curve).

In order to reveal whether or not the previously studied structural model of Gd@C_82_O_3_(OH)_24_ [21] (Figure 2B) correctly represents the atomic structure of experimentally synthesized fullerenols, the corresponding theoretical DFTB3 (green curve) and experimental IR (black curve) spectra are compared in Figure 1. The experimental spectrum of the Gd@C_82_O_x_H_y_ complex showed that the absorption bands provided by the C_82_ carbon cage lay off the bands of gadolinium ion (700–800 cm^−1^). Previously, it was shown that the presence of gadolinium ion slightly affected the carbon cage parameters [22]. Thus, it was assumed that excluding gadolinium ion from the theoretical model does not affect the part of the spectrum provided by the functional groups of the cage but can significantly simplify the calculations. Therefore, the gadolinium was removed from the model of Gd@C_82_O_3_(OH)_24_ (Figure 2C). Comparison of the FTIR experimental and DFTB3 theoretical IR spectra directly demonstrates the good agreement of the spectral lines in the spectral region 900–1400 cm^−1^, which reflects the epoxy and C–O groups’ vibrational modes, with a complete lack of intensity in the energy region (1400–1800 cm^−1^) assigned to the carboxylic and carbonyl groups and double and aromatic carbon–carbon bonds of the C_82_ carbon cage. Atomic coordinates of the C_82_O_3_(OH)_24_ structural model can be found in the Appendix A.

Comparison of the experimental FTIR spectrum with the DFTB3 theoretical one shows that the structural model of C_82_O_3_(OH)_24_ does not satisfy the spectral features of hydroxylated fullerenes. For the C_82_O_3_(OH)_24_ spectral region for C–OH groups and carbon atoms of the cage is 1100–1350 cm^−1^, and for epoxy groups, the spectral region is around 850 cm^−1^ (Figure 1). In contrast to the theoretical spectrum, the experimental FTIR reveals a narrow very intense band of a complex shape in the spectral range from 1600 to 1800 cm^−1^. The spectral shoulder with the energy of 1650 cm^−1^ could be assigned to double C=C bond vibrations, whereas the main peak with the energy of 1750 cm^−1^ could be assigned to the C=O absorption band.

Fullerenes and their derivatives have polyhedral carbon cages in which sp^2^–carbons are directly bonded to three neighbors. Structurally, any fullerene cage cannot bear surface C=O groups without a breakdown of the carbon cage, since it violates the maximum carbon valence of four, and it could be assumed that the Gd@C_82_O_x_H_y_ complex contained functional groups other than epoxy– and hydroxyls. The C=O bond’s presence in Gd@C_82_O_x_H_y_ was confirmed by X–ray photoelectron spectroscopy as well [46]. Following the experimental FTIR spectrum, one can assume novel structural models of Gd@C_82_O_x_H_y_ complexes with open–shell carbon cages caused by excess oxidation by oxygen during high–temperature synthesis.

### 2.2. Atomic Structure of C_82_O_28_H_20_ Valent Isomers

Following the experimental spectroscopic data of synthesized endohedral complexes and previous joint experimental and theoretical results [21], the Gd@C_82_ fullerene (Figure 2A) was considered the basis for a set of initial atomistic structural models. According to the experimental data (see the Experimental Methods Section), the ratio of hydroxyl groups to oxygen is 1:3, with the total number of oxygen atoms approximately equal to 28 [15]. Since the initial Gd@C_82_ fullerene belongs to the C_2v_ group, 28 oxygen atoms were symmetrically distributed over the carbon cage. The oxygen atoms were added to the adjacent carbon atoms of six–membered rings (Figure 2E) and then transformed to hydroxyl, carbonyl and carboxyl functional groups. As a result of the modeling, two isomers of C_82_O_28_H_20_ containing C=O bonds were proposed to explain the peaks of experimental FTIR. The main difference between the isomers is how the C=O bond is coordinated to the C_82_ cage. The fundamental goal of the research is to study the main structural features of Gd@C_82_O_x_H_y_ endohedral complexes and their spectroscopic characteristics.

The first C_82_O_8_(OH)_20_ valent isomer (*I*1) has 20 hydroxyl (−OH) and 8 carbonyl (C=O) groups (Figure 2F) distributed as eight carbonyls and hydroxyl pairs and six pairs of hydroxyls attached to the adjacent carbon atoms. Taking into account the limiting valence of a carbon atom equal to four sp^3^ C–C bonds, one could expect that the oxidation of a carbon atom to carbonyl must proceed with the breakdown of a chemical bond with one of its closest neighbors, with the consequent oxidation of the partner to at least hydroxyl and opening of the fullerene cage between the groups.

The second C_82_O_6_(OOH)_2_(OH)_18_ valent isomer (*I*2) has 18 hydroxyl (−OH), 6 carbonyl (C=O) and 2 carboxyl (−COOH) functional groups (Figure 2G). Two carboxyl groups were placed at opposite sides of the C_82_ cage. Carboxyl groups were formed by a carbon atom lying at the vertex of two neighboring hexagons and a pentagon. Thus, two initial C–C bonds were broken to form the −COOH carboxyl group, and two hydroxyl groups were attached to other ends of each broken C–C bond. Other functional groups were distributed in the same way as in *I*1. Similar to the C_82_O_3_(OH)_24_ model, gadolinium was removed from the new atomistic models. The atomic coordinates of both isomers can be found in the Appendix A.

The atomic structures of both *I*1 and *I*2 isomers optimized at the DFTB3 level of theory are presented in Figure 2H,I. During the DFTB3 optimization, all carbonyl groups of the first isomer (Figure 2F) were transformed into epoxy groups keeping a closed parent C_82_ cage. Thus, the optimized structure of *I*1 has 20 hydroxyl (−OH) and 8 epoxy groups, in which the O atoms bridge the C–C bonds.

The DFTB3 equilibrium atomic structure of the *I*2 isomer is presented in Figure 2I. As a result of optimization, *I*2 has six pores in the cage, with two of them formed by a combination of two hexagons and one pentagon due to the formation of −COOH carboxyl groups. The mutual coordination of an *I*2 carboxyl and two hydroxyl groups prevented the formation of new C–O–C bonds between them with the formation of two epoxy groups. Each “carboxyl pore” has an adjacent pore emerging from the elongation of the distance between C=O and C–OH and combining two hexagons. Two other pores also formed by elongation of the distance between C=O and −OH and combining two hexagons on the C_2v_ axis of the cage. Finally, the *I*2 isomer has an open C_82_ cage with 2 carboxyl, 2 epoxy, 4 carbonyl and 18 hydroxyl groups.

### 2.3. Theoretical IR Spectra of C_82_O_28_H_20_ Valent Isomers

The theoretical IR vibration spectra for both *I*1 and *I*2 isomers calculated at the DFTB3 level of theory are presented in Figure 1 with blue and red curves, respectively. Since both *I*1 and the previously proposed C_82_O_3_(OH)_24_ fullerenole are functionalized by hydroxyl and epoxy groups, one can see the similarity in their theoretical IR spectra, with some differences in the peak intensities caused by different amounts and different coordination of the hydroxyl and epoxy groups at the C_82_ cage. Comparison of the experimental FTIR spectrum (Figure 1) with the DFTB3 theoretical *I*1, one directly demonstrates a good agreement between the spectra in the 850–1350 cm^−1^ spectral region. In contrast, theoretical spectrum does not demonstrate any visible intensity in the carbonyl group spectral region (1750–1780 cm^−1^, Figure 1, blue curve), which is the most intensive region for the experimental one. One can conclude that the IR spectrum of the *I*1 isomer, which is characterized by the presence of eight epoxy groups, fails to describe the main spectral features of the Gd@C_82_O_x_H_y_ complex.

The DFTB3 theoretical IR spectrum of the open−cage *I*2 isomer (Figure 1, red curve) demonstrates a much better agreement with the experimental FTIR one. The presence of the C=O carbonyl and −COOH carboxyl groups has a great impact on the IR spectrum of *I*2 in spectral region 1400–1800 cm^−1^. The DFTB3 calculated IR spectrum has characteristic bands at the 1600–1630 cm^−1^ (carbon cage), 1650–1670 cm^−1^ (vibrations of −COOH groups) and 1690–1720 cm^−1^ (C=O groups) (Figure 1, red curve) spectral regions. The good agreement of the DFTB3 theoretical *I*2 IR spectrum with the experimental FTIR one directly indicates the open−cage nature of the Gd@C_82_O_x_H_y_ complex.

## 3. Methods

### 3.1. Experimental Methods

A mixture of Gd_2_O_3_ powder and graphite in a weighted ratio of 1:1 was processed in high–frequency arc plasma discharge by the sputtering of graphite electrodes with 3 mm axial holes [50]. A fullerene mixture was extracted from the carbon condensate by carbon disulfide in a Soxhlet apparatus. Using the experimental technique described in Reference [51], a mixture of Gd@C_82_ and higher fullerenes was isolated from the resulting solution, and the sample was dried and redissolved in toluene solvent. The Gd@C_82_ endohedral fullerene was isolated from the solution by high–efficiency liquid chromatography on an Agilent Technologies 1200 Series chromatograph. As it was shown [22] in the mass spectrum, only the main fraction of Gd@C_82_ is presented, well isolated from broadband noise. According to the method proposed in Reference [52], the −O and −OH groups were attached to the isolated endohedral metallofullerene. The number of these functional groups was calculated [46] using the X–Ray photoelectron Spectroscopy data from the fraction of carbon atoms chemically bonded to oxygen atoms. Taking into account that the number of −OH hydroxyl groups attached to the fullerene must be even [48], the composition of this product can be presented as Gd@C_82_O_x_(OH)_y_ (x = 10–12, y = 30–32, x + y = 40–42) [46]. The Fourier–transform infrared absorption spectrum (Figure 1, black curve) was recorded using the VERTEX 70 (Bruker Optik GMBH) spectrometer in the spectral region of 400–4000 cm^−1^ with a spectral resolution of 4 cm^−1^. To obtain the spectrum, round tablet samples of 0.5 mm thickness, 13 mm diameter and 0.140 g weight were prepared. The tablets were prepared as follows: Less than 0.00089 g sample weight of Gd@C_82_O_x_H_y_ were thoroughly grounded with 0.14 g of KBr and subjected to cold pressing at 10,000 kg. The FTIR spectrometer was equipped by a global light source, wide band KBr beam splitter and RT–DLaTG detector (Bruker Optic GMBH).

### 3.2. Computational Details

The following computational procedure was used: (1) The fullerenol molecules were designed and constructed, (2) molecules were optimized to an equilibrium geometry and (3) the IR spectrum was calculated. The atomic structure and infrared spectra of free–standing C_82_O_3_(OH)_24_ hydroxylated fullerene and two valent isomers of C_82_O_28_H_20_, namely *I*1 (C_82_O_8_(OH)_20_) and *I*2 (C_82_O_6_(OOH)_2_(OH)_18_), were studied using the Density Functional Tight Binding [53] approach. To achieve high–quality vibrational spectra simulations and strict requirements for the structural optimization of low–dimensional clusters [54], the atomic structures of the Gd@C_82_O_8_(OH)_20_ and Gd@C_82_O_6_(OOH)_2_(OH)_18_ complexes (Figure 2) in the gas phase were optimized with the threshold of 10^−4^ Hartree/Bohr by DFTB3 using 3ob–3–1 parameters [53] using the GAMESS code [55]. Following the mandatory requirements of the Topology Conservation Theorem for low–dimensional lattices [54], the lowering or even the breakdown of high symmetry of low–dimensional atomic lattices due to any kind of internal mechanical stress must be taken into account. Due to this, all electronic structure calculations were performed without taking into account of any kind of point symmetry groups, and the C_1_ symmetry group was applied to all atomistic models of the C_82_ fullerene derivatives.

The DFTB3 is a fast and accurate parametrized quantum–mechanical method suitable to optimize extended molecules and clusters of dozens of atoms containing carbon, oxygen and hydrogen atoms [53] with high accuracy. The DFTB3 method is a very efficient approximation based on DFT parametrization, which provides a reasonably accurate atomic structure and force constants in comparison with the DFT and experimentally measured vibrational spectra [56] for a diverse range of molecules [57].

## 4. Conclusions

Chemically modified C_82_–based fullerenole endohedral complexes with Gd as a guest ion seem to be prospective agents for magnetic resonance imaging and cancer treatment due to the high contrast of Gd in magnetic resonance tomography, inert and protective carbon cages around Gd ions and chemical solubility in a biomedical environment. Gd@C_82_O_x_H_y_ complexes were synthesized and investigated by joint experimental and theoretical techniques combining FTIR spectroscopy and theoretical DFTB3 electronic structure calculations. It was shown that consideration of the IR spectral mechanisms based on just the structure of an oxidized C_82_ cage to consider the force constants at the DFTB3 level of theory without consideration of gadolinium endohedral ion provides fundamental insights into the atomic structure and spectroscopic properties of Gd@C_82_O_x_H_y_ complexes. A comparison of experimental FTIR and theoretical IR spectra of a number of C_82_ fullerenole isomers was performed for the sake of the determination of the atomic structure of a chemically modified Gd@C_82_O_x_H_y_ cage. Based on the proposed theoretical structural models of C_82_O_x_H_y_ fullerenols, theoretical DFTB3 IR spectra were calculated and compared with the experimental FTIR. It was shown that oxidation of the C_82_ cage with the formation of C_82_O_28_H_20_ should lead to the breakdown of some C–C bonds with the integrity breakdown of the parent C_82_ cage with the formation of pores between neighboring carbonyl and carboxyl groups. The Gd@C_82_O_6_(OOH)_2_(OH)_18_ endohedral complex with epoxy, carbonyl and carboxyl groups was considered as the most reliable fullerenole structural model.

## Figures and Tables

**Figure 1 molecules-28-01569-f001:**
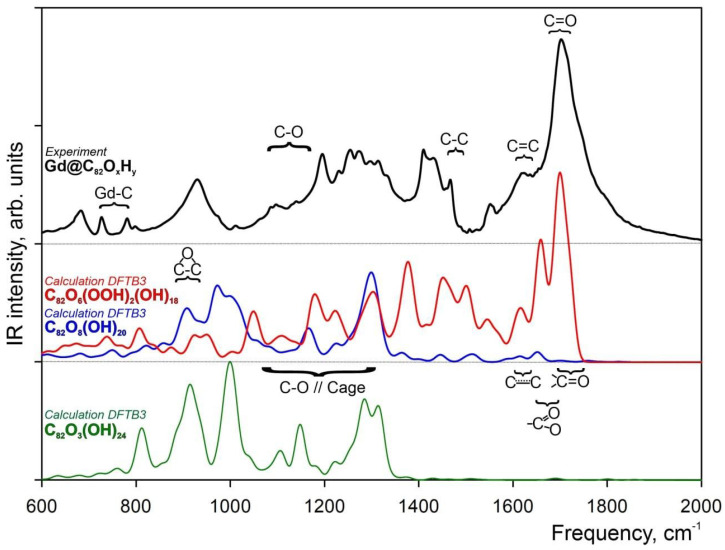
Experimental FTIR spectrum of Gd@C_82_O_x_H_y_ and theoretical spectra of C_82_O_x_H_y_ possible isomers. Top: Experimental FTIR spectrum of Gd@C_82_O_x_H_y_ in the KBr matrix (black curve). Bottom: Theoretical IR spectra of free–standing C_82_O_3_(OH)_24_ (green curve). Middle: Theoretical IR spectrum of *I*1 isomer (C_82_O_8_(OH)_20_, blue curve), and theoretical IR spectrum of *I*2 isomer (C_82_O_6_(OOH)_2_(OH)_18_, red curve). Theoretical IR spectra calculated in the gas phase at the DFTB3 level of theory.

**Figure 2 molecules-28-01569-f002:**
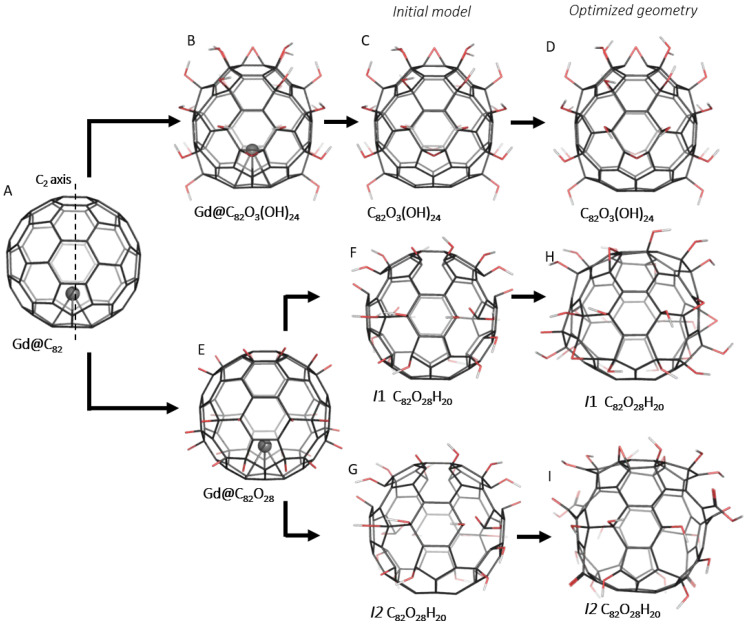
Some structural models of possible C_82_O_x_H_y_ complexes. (**A**) Gd@C_82_; (**B**) Gd@C_82_O_3_(OH)_24_. (**C**,**F**,**G**) Initial structural models of C_82_O_3_(OH)_24_, where *I*1 is C_82_O_8_(OH)_20_ and *I*2 is C_82_O_6_(OOH)_2_(OH)_18_, valent isomers of the C_82_O_28_H_20_ complex, respectively. (**D**,**H**,**I**) Equilibrium geometries of C_82_O_3_(OH)_24_, *I*1 and *I*2 isomers of the C_82_O_28_H_20_ complex, respectively (atomic coordinates of *I*1 and *I*2 isomers can be found in the Appendix A). Carbon, gadolinium, oxygen and hydrogen atoms are depicted in black, gray, red and white, respectively.

## Data Availability

The data that support the findings of this study are available upon reasonable request from the corresponding authors.

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
