# Peer review of "Structure and Vibrational Spectroscopy of C82 Fullerenol Valent Isomers: An Experimental and Theoretical Joint Study"

_molecules, 2023, doi:10.3390/molecules28041569_

Round 1
Reviewer 1 Report
The authors study some C82 fullereneols isomers using theoretical and experimental methods. The comments on this work are listed below:
- The manuscript does not specify if the computed structures are a minimum in the potential electrostatic surface. Please clarify this point.
- There is no theoretical spectrum for the Gd@fulleren system to compare the study
Author Response
Dear Reviewer,
Thank you very much for your kind consideration of the manuscript ID molecules-2155895 of our paper "Structure and Vibrational Spectroscopy of Valent Isomers C82 Fullerenols: an Experimental and Theoretical Joint Study", authored by F.N. Tomilin, P.V. Artyushenko, I.A. Shchugoreva, A.V. Rogova, N.G. Vnukova, G.N. Churilov, N.P. Shestakov, O.N. Tchaikovskaya, S.G. Ovchinnikov, P.V. Avramov, devoted to joint experimental and theoretical study of structure and properties of Gd@С82ОxНy endohedral complexes for advanced biomedical applications.
After careful consideration of your comments we have agreed with all remarks and criticisms and deeply reconsidered the manuscript. All corrections in the text are highlited in red.
Comment 1: The manuscript does not specify if the computed structures are a minimum in the potential electrostatic surface. Please clarify this point.
--- Response: In contrast with GAUSSIAN code, GAMESS program does not calculate a minimum in the potential electrostatic surface. In the study standard GAMESS calculation procedure has been used and a number of fullerenol molecular structures were considered. The geometries of the complexes were optimized and finally the IR spectra were calculated. Corresponding sentence which describes the procedure has been added to Section 3.2 “Computational Details”: “The following computational procedure was used: 1) the fullerenol molecule was designed and constructed; 2) the molecule was optimized to an equilibrium geometry; and 3) the IR spectrum was calculated.”
Comment 2: There is no theoretical spectrum for the Gd@fulleren system to compare the study.
--- Response: The high-temperature synthesis of water-soluble fullerenols (https://doi.org/https://doi.org/10.1016/j.jpcs.2019.109094) is carried out under rather stringent conditions. This leads to a wide range of fullerenol molecules with different numbers and combinations of oxygen-containing functional groups. This in turn leads to difficulties in the determination of the atomic structure by means of X-ray electron diffraction. Therefore, various indirect methods should be and are used to verify the structures, which cannot give a direct and accurate answer about the atomic structure of the fullerenols.
In the study, the FT-IR spectroscopy was used to determine the nature and coordination of the functional groups, since XPS does not provide detailed information about the diversity of functional groups. Certain bands in the experimental FT-IR spectrum suggest that C=O carbonyl, epoxy, -OH hydroxyl, and -COOH carboxyl groups can be formed and coordinated to the surface of fullerenol C82 carbon cage (https://doi.org/10.1021/ja047593o). The atomistic theoretical models of possible C82 fullerenol isomers should take into account that the integrity of the parent fullerene cage could be broken by the appearance of C=O carbonyl and -COOH carboxyl groups coordinated to the surface of the molecule. Therefore, it is important to control that the fullerenol molecule is not break down, i.e., does not disintegrate into different species and fragments, during geometry optimization in electronic structure calculations.
To elucidate the atomic structure of Gd@C82OxHy, comparison of experimental and theoretical IR spectra has been made and the ratio of functional groups has been established based on intensity of corresponding peaks (according to reference [https://doi.org/10.1039/C6DT00223D], absorption bands of C-C (1554 cm-1), C=C (1621 cm-1), C=O (1703 cm-1), and C-O (1050-1150 cm-1) bonds were identified).
The approximate number of the functional groups was determined using XPS based on the proportion of carbon atoms chemically bonded to oxygen. Since the number of -OH hydroxyl groups attached to the fullerene must be even, the composition of a given fullerenol can be expressed as Gd@C82OxHy, where x : y ~ 1.5, so in this work the molecules C82O6(OOH)2(OH)18 and C82O8(OH)20 were constructed with an O : H ratio = 28:20. Since the main comparison of theoretical models and experimental structures was done using the infrared spectroscopy, the main goal in this work is to achieve the correct ratio of functional groups as it appears in the intensity of the corresponding spectral FT-IR peaks. At the same time, gadolinium can be neglected in the calculation of IR spectra, since the concentration of one atom is very small compared to the rest, and, which could be even much more important, due to huge Gadolinium atomic mass the vibrations of the gadolinium ion are in the long wavelength region of the spectrum, all this draw that its characteristic lines are poorly or not visible at all on the experimental spectrum. Therefore, the use of simpler methods such as the effective DFTB3 is more reasonable and faster. Corresponding text has been added to Section 2 “Results and Discussion”.
Best regards,
Prof. P. Avramov
Departmen
Dear Reviewer,
Thank you very much for your kind consideration of the manuscript ID molecules-2155895 of our paper "Structure and Vibrational Spectroscopy of Valent Isomers C82 Fullerenols: an Experimental and Theoretical Joint Study", authored by F.N. Tomilin, P.V. Artyushenko, I.A. Shchugoreva, A.V. Rogova, N.G. Vnukova, G.N. Churilov, N.P. Shestakov, O.N. Tchaikovskaya, S.G. Ovchinnikov, P.V. Avramov, devoted to joint experimental and theoretical study of structure and properties of Gd@С82ОxНy endohedral complexes for advanced biomedical applications.
After careful consideration of your comments we have agreed with all remarks and criticisms and deeply reconsidered the manuscript. All corrections in the text are highlited in red.
Comment 1: The manuscript does not specify if the computed structures are a minimum in the potential electrostatic surface. Please clarify this point.
--- Response: In contrast with GAUSSIAN code, GAMESS program does not calculate a minimum in the potential electrostatic surface. In the study standard GAMESS calculation procedure has been used and a number of fullerenol molecular structures were considered. The geometries of the complexes were optimized and finally the IR spectra were calculated. Corresponding sentence which describes the procedure has been added to Section 3.2 “Computational Details”: “The following computational procedure was used: 1) the fullerenol molecule was designed and constructed; 2) the molecule was optimized to an equilibrium geometry; and 3) the IR spectrum was calculated.”
Comment 2: There is no theoretical spectrum for the Gd@fulleren system to compare the study.
--- Response: The high-temperature synthesis of water-soluble fullerenols (https://doi.org/https://doi.org/10.1016/j.jpcs.2019.109094) is carried out under rather stringent conditions. This leads to a wide range of fullerenol molecules with different numbers and combinations of oxygen-containing functional groups. This in turn leads to difficulties in the determination of the atomic structure by means of X-ray electron diffraction. Therefore, various indirect methods should be and are used to verify the structures, which cannot give a direct and accurate answer about the atomic structure of the fullerenols.
In the study, the FT-IR spectroscopy was used to determine the nature and coordination of the functional groups, since XPS does not provide detailed information about the diversity of functional groups. Certain bands in the experimental FT-IR spectrum suggest that C=O carbonyl, epoxy, -OH hydroxyl, and -COOH carboxyl groups can be formed and coordinated to the surface of fullerenol C82 carbon cage (https://doi.org/10.1021/ja047593o). The atomistic theoretical models of possible C82 fullerenol isomers should take into account that the integrity of the parent fullerene cage could be broken by the appearance of C=O carbonyl and -COOH carboxyl groups coordinated to the surface of the molecule. Therefore, it is important to control that the fullerenol molecule is not break down, i.e., does not disintegrate into different species and fragments, during geometry optimization in electronic structure calculations.
To elucidate the atomic structure of Gd@C82OxHy, comparison of experimental and theoretical IR spectra has been made and the ratio of functional groups has been established based on intensity of corresponding peaks (according to reference [https://doi.org/10.1039/C6DT00223D], absorption bands of C-C (1554 cm-1), C=C (1621 cm-1), C=O (1703 cm-1), and C-O (1050-1150 cm-1) bonds were identified).
The approximate number of the functional groups was determined using XPS based on the proportion of carbon atoms chemically bonded to oxygen. Since the number of -OH hydroxyl groups attached to the fullerene must be even, the composition of a given fullerenol can be expressed as Gd@C82OxHy, where x : y ~ 1.5, so in this work the molecules C82O6(OOH)2(OH)18 and C82O8(OH)20 were constructed with an O : H ratio = 28:20. Since the main comparison of theoretical models and experimental structures was done using the infrared spectroscopy, the main goal in this work is to achieve the correct ratio of functional groups as it appears in the intensity of the corresponding spectral FT-IR peaks. At the same time, gadolinium can be neglected in the calculation of IR spectra, since the concentration of one atom is very small compared to the rest, and, which could be even much more important, due to huge Gadolinium atomic mass the vibrations of the gadolinium ion are in the long wavelength region of the spectrum, all this draw that its characteristic lines are poorly or not visible at all on the experimental spectrum. Therefore, the use of simpler methods such as the effective DFTB3 is more reasonable and faster. Corresponding text has been added to Section 2 “Results and Discussion”.
Best regards,
Prof. P. Avramov
Department of Chemistry, College of Natural Sciences,
Kyungpook National University,
80 Daehak-ro, Buk-gu, Daegu, 41566, Republic Korea
t of Chemistry, College of Natural Sciences,
Kyungpook National University,
80 Daehak-ro, Buk-gu, Daegu, 41566, Republic Korea

Reviewer 2 Report
This work combines theoretical DFTB3 analysis with actual Fourier transform infrared (FT-IR) absorption spectroscopy to better understand the intricate structure of the Gd@C82OxHy. The results revealed that the endohedral Gd@C82O6(OOH)2(OH)18 combination containing epoxy, carbonyl, and carboxyl groups was the most trustworthy fullerenole structural mode. This manuscript was well organized and the conclusions can be supported by the data. The following points should be addressed before publication:
1. Please state the innovation of this manuscript.
2. To acquire the spectra simulations and structure, the DFTB3 was selected. By contrasting numerous other approaches, the author should show that it is a better option.
3. English writing should be polished and improved. For example, Gd@C82 fullerene should be Gd@C82 in line 123. 400÷4000 cm-1 should be 400-4000 cm-1 in line 203.
4. The open-cage I2 isomer's IR spectra and experimental FT-IR results coincide, and the author believes the complex is composed of Gd@C82O6(OOH)2(OH)18. To further verify the structure, another characterization technique should be used.
5. The two isomers of C82O6(OOH)2(OH)18 and C82O8(OH)20 were taken into consideration. Please explain clearly why and what basis these two monomers were selected.
6. In order to simplify the computations and account for the slightly altered carbon cage properties, gadolinium was removed. Please list the works of literature that employ this method.
7. The interaction between guest and host of endohedral fullerenes can be further addressed with referenced to previous reports, such as:
10.3390/nano9040630; 10.1016/j.physe.2020.114532
Author Response
Dear Reviewer,
Thank you very much for your kind consideration of the manuscript ID molecules-2155895 of our paper "Structure and Vibrational Spectroscopy of Valent Isomers C82 Fullerenols: an Experimental and Theoretical Joint Study", authored by F.N. Tomilin, P.V. Artyushenko, I.A. Shchugoreva, A.V. Rogova, N.G. Vnukova, G.N. Churilov, N.P. Shestakov, O.N. Tchaikovskaya, S.G. Ovchinnikov, P.V. Avramov, devoted to joint experimental and theoretical study of structure and properties of Gd@С82ОxНy endohedral complexes for advanced biomedical applications.
After careful consideration of your comments we have agreed with all remarks and criticisms and deeply reconsidered the manuscript. All corrections in the text are highlited in red.
Reviewer 2
Comment 1: Please state the innovation of this manuscript.
--- Response: Innovation of the study can be expressed following 3 points:
1) Atomic structure of synthesized C82O28H20 has not been properly studied before using either experimental or theoretical approaches.
2) In contrast to previous theoretical studies of hydroxylated derivatives of C82, this study represents a joint experimental and theoretical approach based on comparison of experimental FT-IR spectra of fullerene derivatives with theoretical DFTB3 IR spectra. A corresponding text was added to Introduction Section: “Although considerable research has been devoted to the medical application of the Gd@C82 derivatives (10.1039/c2ib20145c, 10.1088/1468-6996/12/4/044607, doi.org/10.1002/viw2.7), rather less attention has been paid to the determination of the atomic structure of functionalized Gd@C82 fullerene (doi:10.1080/00268976.2011.591743, 10.3390/computation9050058, doi.org/10.1016/j.cplett.2010.04.007). At the moment, the description of the atomic structure of fullerene was based on the prediction of the stability of various models by the methods of density functional theory without comparison with experimental data (doi:10.1080/00268976.2011.591743, 10.3390/computation9050058, doi.org/10.1016/j.cplett.2010.04.007).”
3) For the first time the open-cage models of C82 cage functionalized by C=O carbonyl, epoxy, -OH hydroxyl, and -COOH carboxyl groups were proposed and studied using joint experimental FT-IR and DFTB3 theoretical approaches.
Comment 2: To acquire the spectra simulations and structure, the DFTB3 was selected. By contrasting numerous other approaches, the author should show that it is a better option.
--- Response: The DFTB3 approach is a fast and accurate quantum-chemical method. It is suitable to optimize molecules and clusters build up by carbon, oxygen, and hydrogen atoms (https://doi.org/10.1021/ct300849w) with high accuracy. DFTB method is very efficient approximation based on DFT parametrization which provides reasonably accurate atomic structure and force constants in comparison with DFT and experimentally measured vibrational spectra (https://doi.org/10.1063/1.4966918) for diverse range of molecules (https://doi.org/https://doi.org/10.1002/jcc.25390). It was shown (https://doi.org/10.1063/1.4966918, https://doi.org/10.1002/jcc.25390), that the DFTB3 method is very effective for the describing of IR spectra in comparison with various DFT potentials and experimental FT-IR spectra. In particular, in the works cited above, a solid comparative study of DFTB approach using a wide set of molecules is presented.
Comment 3: English writing should be polished and improved. …
--- Response: We deeply reconsidered the text of the manuscript and in particular carefully polished and improved English language beginning the Title of the manuscript with the new title of the paper: “Structure and Vibrational Spectroscopy of C82 Fullerenol Valent Isomers: an Experimental and Theoretical Joint Study”. Also, English grammar has been improved and misprints have been.
Comments 4, 5, and 6:
- The open-cage I2 isomer's IR spectra and experimental FT-IR results coincide, and the author believes the complex is composed of Gd@C82O6(OOH)2(OH)18. To further verify the structure, another characterization technique should be used.
- The two isomers of C82O6(OOH)2(OH)18 and C82O8(OH)20 were taken into consideration. Please explain clearly why and what basis these two monomers were selected.
- In order to simplify the computations and account for the slightly altered carbon cage properties, gadolinium was removed. Please list the works of literature that employ this method.
--- Joint response for comments 4, 5, and 6: Joint response for comments 4, 5, and 6 has been maid because they have a common answer: The following text has been to Results and Discussion Section:
The high-temperature synthesis of water-soluble fullerenols (https://doi.org/https://doi.org/10.1016/j.jpcs.2019.109094) is carried out under rather stringent conditions. This leads to a wide range of fullerenol molecules with different numbers and combinations of oxygen-containing functional groups. This in turn leads to difficulties in the determination of the atomic structure by means of X-ray electron diffraction. Therefore, various indirect methods should be and are used to verify the structures, which cannot give a direct and accurate answer about the atomic structure of the fullerenols.
It is reasonable to use the infrared spectroscopy and X-ray photoelectron spectroscopy (XPS) to certify the water-soluble fullerenes. The FT-IR spectroscopy method allows one to determine the functional groups, while XPS provides the opportunity to calculate the number of bond species of carbon and oxygen atoms. In the XPS fullerenol spectrum, the C1s bond energies extend to 5.3 eV and clearly deviate from the C1s peak of fullerene (https://doi.org/10.1021/ja047593o). The large shift indicates a relatively high percentage of carbon with a higher degree of oxidation in fullerenol compared to fullerene. The wide range and spectral width also directly indicates that a significant number of isomers may be presented in the experimental mixture of the products. Analysis of the data provides unique possibility to calculate the approximate atomic concentrations of different carbon and oxygen species with different bond types and oxidative states.
In the study, the FT-IR spectroscopy was used to determine the nature and coordination of the functional groups, since XPS does not provide detailed information about the diversity of functional groups. Certain bands in the experimental FT-IR spectrum suggest that C=O carbonyl, epoxy, -OH hydroxyl, and -COOH carboxyl groups can be formed and coordinated to the surface of fullerenol C82 carbon cage (https://doi.org/10.1021/ja047593o, https://doi.org/https://doi.org/10.1016/S0166-1280(01)00756-4, https://doi.org/10.1039/C6DT00223D). The atomistic theoretical models of possible C82 fullerenol isomers should take into account that the integrity of the parent fullerene cage could be broken by the appearance of C=O carbonyl and -COOH carboxyl groups coordinated to the surface of the molecule. Therefore, it is important to control that the fullerenol molecule is not break down, i.e., does not disintegrate into different species and fragments, during geometry optimization in electronic structure calculations.
The approximate number of the functional groups was determined using XPS based on the proportion of carbon atoms chemically bonded to oxygen. Since the number of -OH hydroxyl groups attached to the fullerene must be even, the composition of a given fullerenol can be expressed as Gd@C82OxHy, where x : y ~ 1.5, so in this work the molecules C82O6(OOH)2(OH)18 and C82O8(OH)20 were constructed with an O : H ratio = 28:20. Since the main comparison of theoretical models and experimental structures was done using the infrared spectroscopy, the main goal in this work is to achieve the correct ratio of functional groups as it appears in the intensity of the corresponding spectral FT-IR peaks. At the same time, gadolinium can be neglected in the calculation of IR spectra, since the concentration of one atom is very small compared to the rest, and, which could be even much more important, due to huge Gadolinium atomic mass the vibrations of the gadolinium ion are in the long wavelength region of the spectrum, all this draw that its characteristic lines are poorly or not visible at all on the experimental spectrum. Therefore, the use of simpler methods such as the effective DFTB3 is more reasonable and faster.
Comment 7: The interaction between guest and host of endohedral fullerenes can be further addressed with referenced to previous reports, such as: 10.3390/nano9040630; 10.1016/j.physe.2020.114532
--- Response: Thank you very much for the valuable data. Information from these papers has been added to Introduction Section: “Density Functional Theory (DFT) calculations is a powerful tool to describe atomic and electronic structure and stability of different derivatives of fullerenes (DOI: 10.1021/acs.jpcc.6b06484, DOI: 10.1021/acs.jpca.8b06340, 10.1007/s11224-012-0137-5, doi.org/10.1016/j.saa.2014.08.023) including endohedral metallofullerenes (doi.org/10.3390/nano9040630, 10.1016/j.physe.2020.114532, doi: 10.1063/1.4904389, doi.org/10.1016/j.comptc.2022.113878, DOI:10.5402/2012/208234, doi.org/10.1021/nl049164u).”
Best regards,
Prof. P. Avramov
Department of Chemistry, College of Natural Sciences,
Kyungpook National University,
80 Daehak-ro, Buk-gu, Daegu, 41566, Republic Korea

Round 2
Reviewer 2 Report
I think the paper can be accepted in present form